# Taming the Noisy Oracle: Robust Entity-Centric Question Answering via Learning from Imperfect Feedback

## Abstract

Entity-centric question answering (ECQA) is the problem of selecting which entities from a large, predefined set are most relevant to given observations. This task highlights a critical challenge for robust machine learning: reliably extracting factual knowledge from LLMs when they are treated as imperfect, black-box information sources, especially with long, heterogeneous inputs. For example, given genes active in a disease, scientists want to identify which biological processes are involved, a task demanding high reliability. Current approaches attempt to achieve robustness through consensus ranking or iterative validation, but these methods incur "token explosion," where costs scale poorly, making them impractical.

We introduce ARISE (Adaptive Residual Information Sampling Engine), a framework that reframes ECQA as a problem of sequential decision-making under structured, imperfect feedback. Our key insight is that each query provides a form of biased data: noisy side-observations about related entities. We leverage this insight with DUETS Bandit (DUal Experts for Turbid side-Observations with Stochastic feedback graph), a novel online learning algorithm designed for this setting. DUETS employs dual expert advisors to navigate this uncertainty: a GraphExpert that models prior knowledge as a stochastic feedback graph to handle data biases, and a NoiseExpert that strategically queries the LLM to maximize observation quality, while Confirmation Atoms validate outputs to update internal beliefs in this interactive environment. This architecture enables statistically rigorous hypothesis testing with formal p-values, creating a robust and reliable system that dramatically reduces query complexity. Preliminary results on synthetic data are promising, and we are currently evaluating ARISE on the challenge of pathway enrichment analysis using 180+ annotated gene expression datasets, a domain where robustness to distribution shift (novel experimental data) is paramount.

## 1 Introduction

Large Language Model (LLM)-based question answering has emerged as a highly active research area. Within this field, we investigate a constrained yet critical paradigm: prompt-only entity-centric question answering (ECQA). Here, the prompt is a self-contained knowledge base, and the LLM must classify which predefined target entities are relevant to the provided observables. This paradigm, however, directly confronts the challenge of learning with imperfect data, as the LLM itself acts as a noisy and often unreliable information oracle. Its outputs are subject to well-documented failure modes like hallucination and factual inconsistency Huang et al. [2024], Wang et al. [2024], which compromise the reliability of any downstream task.

These challenges are amplified in scientific discovery, where queries often involve complex inputs and demand high-confidence answers. A scientist might query an LLM based on novel laboratory results,

which introduces a significant distribution shift, as the query is conditioned on out-of-distribution scientific data not present in the LLM's training set. A quintessential example is Pathway Enrichment Analysis (PEA), a specific instance of ECQA where target entities are biological pathways and observables are gene lists from experiments. Scientists seek to answer: "What is the underlying functional meaning of these genes?" This question is central to bioinformatics, and its difficulty highlights the need for robust methods that can handle novel, noisy, and complex biological data.

Existing methods attempt to bolster LLM safety and alignment with facts through several strategies: 1) consensus aggregation from partial queries [Singhal, 2025, Wang et al., 2023a]; 2) confidence scoring to manage uncertainty Hüllermeier and Waegeman [2021], Zong and Huang [2025]; and 3) agentic, web-enabled architectures to handle out-of-distribution queries Gao et al. [2024], Xi et al. [2023]. Despite these advances, a harsh trade-off between robustness and computational cost remains. Achieving high reliability often requires iterative feedback loops that are computationally infeasible for large-scale problems Chen et al. [2024].

We directly address this cost-robustness trade-off by designing a system that makes principled decisions under uncertainty. Our approach is built on three key insights for handling imperfect information from LLMs: First, each query provides biased and partial information about all entities, not just the one targeted. Second, prior knowledge about entity co-occurrence can be used to model and mitigate this bias. Third, residual information from the validation process itself can be used to refine future decisions.

To this end, we introduce ARISE (Adaptive Residual Information Sampling Engine), a framework that provides a statistically-grounded orchestration for iterative retrieval. ARISE is built from two symbiotic parts: a smart sampling policy that learns to manage biased data from both prior knowledge and online LLM feedback, and a statistical engine that enables online validation by formulating an appropriate null distribution.

At the heart of ARISE is DUETS Bandit ("DUal Experts for Turbid side-Observations with Stochastic feedback graph"), a novel multi-armed bandit algorithm. DUETS models the problem as an "expert" setting where each action (query) reveals a corrupted signal about all outcomes. It features a GraphExpert that leverages prior knowledge as a stochastic feedback graph to counteract biased sampling Mannor and Shamir [2011], Alon et al. [2017], and a NoiseExpert that strategically selects queries to maximize the quality of the LLM's noisy feedback. By adaptively mixing their advice, DUETS achieves a highly efficient and robust sampling scheme.

The rest of the paper is structured as follows: Section 2 positions our work relative to ECQA and online learning for robustness. Section 3 details the ARISE framework. Finally, Section 4 presents our evaluation and discusses ongoing work.

## 2 Related Works and Positioning

Zero-Shot Entity-Centric Question Answering (which we refer here simply as ECQA) is characterized by several key exclusions. It operates without Retrieval-Augmented Generation (RAG) [Lewis et al., 2020], fine-tuning, or access to the model's output probabilities. Consequently, the model's weights are frozen, its reasoning is confined to its in-context learning abilities (including MCP Hou et al. [2025]), and it is treated as a black box.

A defining feature of our ECQA setup is the complexity of the input, which directly confronts a primary architectural limitation of modern LLMs: the effective utilization of long, information-dense, and multimodal context windows. While new models feature massive context windows, research demonstrates a significant gap between this theoretical capacity and practical reasoning ability, effects like "lost in the middle" [Liu et al., 2023], hallucinations [Huang et al., 2024], or "long-tail knowledge collapse" Kandpal et al. [2023], are well-documented and results in sharp performance decay. This performance decay is not merely theoretical, for a task like PEA, a long list of input genes can cause a diagnostically critical gene to be effectively ignored if it falls into this neglected middle section [Liu et al., 2023, Shi et al., 2024, Yuan et al., 2024]. The model's subsequent reasoning is thus based on a flawed and incomplete representation of the input, leading to an incorrect classification. This failure stems not from a lack of knowledge but from an architectural artifact of processing long sequences [Shi et al., 2024].

To overcome these constraints, prompt engineering has become a leading strategy [Liu et al., 2023]. Effective prompts often mimic domain-specific reasoning patterns, analogous to Chain-of-Thought

[Wei et al., 2022]. A prime example in bioinformatics is the TALISMAN method, which explicitly instructs the model to perform a "term enrichment test" on a list of genes, forcing it to synthesize a high-level biological concept [Yuan et al., 2024]. Similarly, in medical diagnosis, a two-step prompt that first organizes clinical data before deriving a diagnosis [Singhal et al., 2023]. Here we address those methods as *"confirmation processes"*, and incorporate them into our framework.

Another line of work develops a more robust architectural pattern of partition-query-aggregate Liu et al. [2025]. These approaches decompose the long, heterogeneous list of observations into smaller partitions, query the LLM on each one, and then synthesize the final result based on the framework of Consensus Ranking from Partial Observations Kemeny and Snell [1962]. While very effective, these architectures come with an extremely high computational cost Wang et al. [2023b], Simeoni et al. [2024], requiring numerous LLM calls. Hence, current research is focused on optimizing parts of the architecture, from context-aware approaches for observation partitioning such as semantic partitioning using feature clustering Saito et al. [2025] , or agentic partitioning Wu et al. [2025], to faster weighted Consensus Ranking algorithms Wang et al. [2025].

Pathway Enrichment Analysis (PEA) is a widely studied field Nguyen et al. [2019], Reimand et al. [2019], Mathur et al. [2018] with extensive validation efforts Geistlinger et al. [2021], Buzzao et al. [2024] , yet it faces several well-documented limitations Lazareva et al. [2021], Khatri et al. [2012], Mubeen et al. [2022] . These limitations often arise from the difficulty of establishing a singular, comprehensive knowledge base, as the required biological knowledge is constantly updating, profoundly heterogeneous, and context-dependent Kotrys et al. [2024], Mubeen et al. [2022]. Those challenges have led to massive collaborative efforts by dedicated human task forces to manually curate biological information from the literature, epitomized by resources like the Kyoto Encyclopedia of Genes and Genomes (KEGG) database Kanehisa and Goto [2000], Kanehisa et al. [2023]. **Those efforts highlights the immense promise of leveraging LLMs for this task, given their potential for deep biological understanding and their capacity to integrate real-time knowledge.** Unfortunately, attempting to apply LLMs directly to this problem often falls short Hu et al. [2025a, 2023], as the specific difficulties of LLM-based PEA are a clear manifestation of the general ECQA challenges previously discussed.

## 2.1 Online Learning with Side-Information

Our framework is a novel application within the broader field of sequential decision-making, which evolved from the seminal frameworks of prediction with expert advice Cesa-Bianchi and Lugosi [2006], where the learner observes the loss of all possible actions at each step (also known as the "full-information" or "expert" setting), and the classic Multi-Armed Bandit (MAB) problem Robbins and Monro [1951], where the learner only observes the loss of the single action they chose (also known as the "bandit" setting).

Here, we focus on a middle ground where side-information for every chosen action exists, meaning choosing one action reveals partial information about others. Specifically, our work incorporates and synthesizes two distinct fields: 1) The **graph-structured feedback model**, introduced by Mannor and Shamir [2011] and extensively developed by Alon et al. [2017]. This framework formalizes side-information using a feedback graph where an edge from action i to j means playing i reveals the loss of j. Key distinctions in this literature include the **informed setting**, where the learner knows the feedback graph before choosing an action, versus the **uninformed setting**. Further nuances involve whether the graph is **symmetric** (reciprocal feedback) or **directed**, and whether it is fixed or **time-varying** Alon et al. [2017]. The work of Li et al. [2019] extends this framework to stochastic graphs where each edge is associated with a probability of being realized. 2) **Learning with noisy side observations** Kocák et al. [2016]. This framework models a different form of side partial information. Instead of the feedback's existence being sparse, it is assumed to be fully present but corrupted by noise.

## 3 Methodological Rationale and Core Components

At the heart of ARISE is a conceptualization of the entity identification problem as a Multi-Armed Bandit (MAB) task. Each candidate entity is formally treated as an arm of the bandit, and the action of pulling an arm constitutes a complete, multi-stage investigative cycle. This cycle begins with query formulation and execution, where a query is constructed by sampling a representative subset of observables from the entity-observable joint distribution, according to the framework's current beliefs, and is then executed against the LLM. The LLM's response is then subjected to a multi-stage

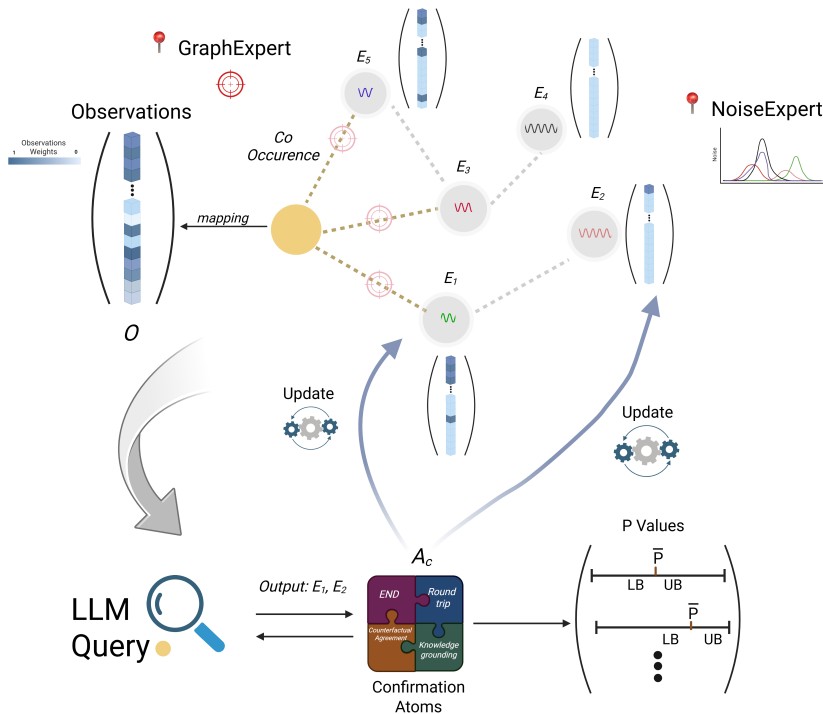

Figure 1: Overview of the **ARISE** framework with its dual-expert online learning algorithm **DUETS**. Observations $O$ (e.g., gene expression features, colored by their weights in the observations' vector) are mapped to candidate entities $E_i$. The **GraphExpert** leverages co-occurrence priors via a feedback graph, while the **NoiseExpert** evaluates the quality of observations across all entities. Outputs from the LLM queries ($E_1$, $E_2$) are validated through a modular system of **Confirmation Atoms** ($A_c$), which assess different sources of uncertainty. Their residual information updates both the statistical significance engine (p-values with confidence intervals) and the experts' internal states, enabling an adaptive and efficient entity-centric question answering pipeline.

validation protocol via a modular suite of Confirmation Atoms, which assess the output's stability, coherence, and factual grounding to return a quantitative confidence score. This validation process also yields residual information that is immediately leveraged to perform online updates to the framework's internal belief structures. Subsequently, the confidence-weighted outcome is assimilated by a Statistical Significance Engine that aggregates evidence across multiple trials against an explicit null hypothesis, culminating in a p-value and a confidence interval to quantify the significance of each entity's observation. Finally, if an entity is considered "statistically enriched" (either positively or negatively), it is masked from subsequent rounds. The intelligent orchestration of this cycle is managed by the DUETS (DUal Experts for Turbid side-Observations with Stochastic feedback graph) bandit algorithm. Figure 1 presents a conceptual overview of the framework.

## 3.1 Generative Model and Statistical Components

As described before, we assume some reference corpus exists of the relation between entities and observables, and Supplementary Section D discusses the case where this data is absent.

**Mapping Observables to Entities**   We model the generation of a set of observables $g_q$ as a draw from a mixture model, where each component corresponds to an entity $E_i$. Each entity $E_i$ is characterized by a categorical distribution over the universe of $N_{\text{back}}$ observables, $\mathcal{O}$. The parameters of this distribution, a probability vector $\vec{\theta_i} \in \Delta^{N_{\text{back}}-1}$, are assumed to be drawn from a conjugate Dirichlet prior, governed by a concentration parameter vector $\vec{\alpha}_i$, and this constitutes a Dirichlet-Multinomial (D-M) model.

The posterior Dirichlet parameters, $\vec{\alpha}'$, are learned from a reference corpus built from a set of datasets, each corresponding to a ranked list of all observables and a set of observed entities. The ranking is

167 based on the assumption that observables with a higher rank are more strongly associated with at least
168 one of the entities. These ranked lists are partitioned into $m$ quintiles, with each quintile assigned a
169 distinct, monotonically decreasing weight. The weights for each entity are then aggregated across the
170 corpus to form an empirical count vector, $\vec{C}_i$.

**Modeling and Updating Entity Relationships** For leveraging the relationships between entities,
172 which form the basis of the stochastic structured feedback graph described in 3.2, we need to ensure
173 the modeled relationships are relevant for propagating the residual loss and enabling updates from
174 the auxiliary information provided by the confirmation atoms. The stochastic feedback graph is the
175 graph in which entities are the nodes, and the edges are the conditional probability of observing
176 entity $E_j$ given the presence of entity $E_i$, denoted $P(E_j|E_i)$. While this probability can be estimated
177 directly from co-occurrence frequencies via the MLE, such an approach is often brittle, especially
178 with small sparse data. We instead employ a Bayesian methodology that provides regularization,
179 robustly handles unseen events, and allows for efficient, sequential updates.

180 We model the conditional probability $P(E_j|E_i)$ as a latent parameter $\theta_{j|i} \in [0, 1]$. For a given entity
181 $E_i$, the presence or absence of any other entity $E_j$ in the same dataset is treated as a Bernoulli
182 trial. To facilitate Bayesian inference, we place a conjugate *Beta* prior on this parameter: $\theta_{j|i} \sim$
183 Beta$(\alpha_{j|i}, \beta_{j|i})$. A weakly informative prior (e.g., $\alpha_{j|i} = 1, \beta_{j|i} = 1$) is chosen to regularize the
184 estimate while allowing the data to drive the posterior.

Given corpus-wide counts of entity occurrences ($N_i$) and co-occurrences ($N_{i,j}$), the posterior distribution for the parameter is also a Beta distribution, $\theta_{j|i}|\text{data} \sim \text{Beta}(\alpha'_{j|i}, \beta'_{j|i})$, with updated parameters: $\alpha'_{j|i} = \alpha_{j|i} + N_{i,j}$, and $\beta'_{j|i} = \beta_{j|i} + (N_i - N_{i,j})$. Then, the point estimate for the conditional probability is the mean of this posterior :

$$P(E_j|E_i) = \frac{\alpha'_{j|i}}{\alpha'_{j|i} + \beta'_{j|i}} = \frac{\alpha_{j|i} + N_{i,j}}{\alpha_{j|i} + \beta_{j|i} + N_i}$$

185 This Bayesian approach offers significant advantages over the MLE ($P(E_j|E_i) = N_{i,j}/N_i$). The
186 prior acts as a smoothing mechanism, preventing the model from assigning probabilities of exactly 0 or
187 1 based on limited observations (the "zero-frequency problem"), which ensures more robust estimates
188 in sparse data regimes. Furthermore, the model is inherently updatable. New data, summarized by
189 counts $N'_i$ and $N'_{i,j}$, can be incorporated by treating the current posterior parameters ($\alpha'_{j|i}, \beta'_{j|i}$) as
190 the new prior and applying the same update rules, avoiding the need to reprocess the entire corpus.

**The Statistical Significance Engine** For a grounded result, we need a mechanism to aggregate
192 iterative queries until a true signal emerges. We achieves this by formal statistical confidence,
193 providing p-value for each entity. For that, we **explicitly build the null hypothesis** ($H_0$), which
194 defined as the probability of observing an entity given the prior beliefs only, position our framework as
195 an "enrichment over current belief" enrichment problem. As described before, Supplementary Section
196 D discuses the case where no prior belief is given and the enrichment is defined over background
197 noise.

A central challenge is that our framework is built on sequential querying over sampled sub-sets,
which are intentionally biased through the prior beliefs of the played action, meaning the probability
of observing an entity changes with every trial. The correct underlying model is therefore a *Poisson
Binomial distribution*, where the prior beliefs probabilities are:

$$P(E_i = 1|g_q) = \frac{P(g_q|E_i) \cdot \pi_i}{P(g_q|E_i) \cdot \pi_i + P(g_q|\neg E_i) \cdot (1 - \pi_i)}$$

198 Where $g_q$ is the current queried set of observables, $\pi_i = P(E_i = 1)$ is the prior probability for each
199 entity being observed, and $P(g_q|\neg E_i)$ is the observables probability for the "background". In our
200 current "working example" where a reference corpus exists, we can easily infer $\pi_i$ and $P(g_q|\neg E_i)$
201 from the data. Supplementary Section D discusses the case where those not exists.

202 For a given entity $E_i$, let $X$ be the random variable for its total count across $T$ trials, and let $k$ be
203 the observed count. Under the null hypothesis, $X$ follows a Poisson Binomial distribution defined
204 by the set of success probabilities $\{p_i(g_{q(1)}), \ldots, p_i(g_{q(T)})\}$. Since we are testing for enrichment,
205 we perform a one-tailed test. The p-value is the probability of observing a count of $k$ or greater

by chance :p-value $= P(X \geq k) = \sum_{j=k}^{T} P(X = j)$. Directly computing the probability mass function $P(X = j)$ is computationally infeasible as it requires summing over an exponential number of combinations, but efficient methods exists Biscarri et al. [2018].

Our framework requires the incorporation of two origins of uncertainty for a robust confidence assessment. The first is the *sampling variance*, for ensuring robustness across any number of trials. The second is the *observation variance*, returned from the confirmation atoms, which reflects the certainty associated with each individual query results. For this, we construct a confidence interval for the empirical success probability. Given confidence interval for the p-value estimator itself is also not analytically feasible, we leverage the duality between hypothesis tests and confidence intervals: Rather than framing the confidence on the p-value, we construct a CI for the empirical success probability parameter $\hat{p}$, with this CI incorporating both the origins of uncertainty. Given it is critical to be robust for any number of trials, we build upon the Clopper-Pearson(C-P) method for the sampling variance CI, and MCMC with adaptive stopping for incorporating the observation variance into this CI.

Specifically, we treat the confidence from each observation as its probability of being a true positive, $P(\text{True observation}|E_i = 1)$, and in each iteration, we sample an "effective k" from the resulting distribution. A C-P interval is calculated for this simulated count, generating a distribution of plausible lower and upper bounds. To construct a single CI which accounts for both sources of uncertainty simultaneously, we use the simulation to derive a confidence interval on the bounds themselves; the final lower bound is taken from the lower tail of the distribution of simulated lower bounds, and the final upper bound from the upper tail of the distribution of simulated upper bounds. An entity is considered "enriched" only if its p-value is below a significance threshold **and** its prior probability, $\pi_i$, falls outside this composite confidence interval.

## 3.2 The Arm Selection Policy

The motivation for our arm selection policy is to intelligently reconcile two distinct beliefs about the data, informed by prior literature and our Confirmation Atoms (CA). The first belief is the co-occurrence probability between entities, which we model as a probabilistic feedback graph to guide exploration. The second is the mapping between observables and entities, which dictates the relevance of information we expect to receive from each query. Our 'DUETS Bandit'(or simply 'DUETS') algorithm is designed to synthesize these two beliefs while accounting for the framework's inherently biased query mechanism; by using observables sampled for one entity to query the LLM about all entities, we receive a turbid signal for each entity.

To achieve this, the core of the 'DUETS' algorithm is its unique dual-perspective architecture. It maintains two parallel expert advisors, each operating under a different worldview, and learns to synthesize their advice. The **'GraphExpert'** is designed to enforce the co-occurrence prior. It operates as if it were in the informed, partial-information setting of Alon et al. [2017], and more specifically under the stochastic setup of Li et al. [2019], treating the realized co-occurrence graph $G_t$ as a feedback mechanism. By focusing its exploration strategy on structurally important nodes (e.g., a dominating set), it ensures that the sampling policy take into account the known relationships between entities.

The **'NoiseExpert'** acknowledges the noisy full-information reality of the problem, resamples the noisy side-observation model of Kocák et al. [2016]. Its goal is to strategically select the query (action) that is *expected* to yield the highest quality information across all entities. It does this by performing a proactive lookahead calculation, using a learned model of observation quality to identify the most informative query to make in each round. This lookahead function is intuitively defined as:

$$\hat{p}_g(i,j) = E_{o \sim P(\cdot|E_i)}[P(E_j|o)] \tag{1}$$

Which is the expected posterior probability of entity j, where the expectation is taken over all the input observables that a query for entity i is likely to produce. Direct computation of this expectation is analytically intractable, we therefore propose an approximation. Given that Equation 1 represents the confusability between entities $E_i$ and $E_j$, an intuitive and computationally efficient solution is to define a score based on the information-theoretic similarity of the entities' learned distributions. Specifically, the Kullback-Leibler (KL) divergence between their posterior Dirichlet distributions, $D_{KL}(\text{Dir}(\vec{\alpha}'_i)||\text{Dir}(\vec{\alpha}'_j))$, measures the inefficiency of using the distribution of $E_j$ to describe observables generated from $E_i$. Supplementary Section B discusses the theoretical

justifications beyond the score. We leverage this by defining a similarity score via an exponential kernel, which serves as a principled proxy for the desired expectation:

$$\hat{p}_g(i,j) := \exp\left(-D_{KL}(\text{Dir}(\vec{\alpha}'_i)\|\text{Dir}(\vec{\alpha}'_j))\right) \tag{2}$$

This score provides a fast and robust measure of entity similarity, directly grounded in the information content of their learned models, which we use in place of the intractable expectation.

'DUETS' then uses a high-level **'Meta-Expert'** that adaptively learns how to best mix the recommendations from these two distinct advisors. By tracking the historical performance of the 'GraphExpert''s structural advice and the 'NoiseExpert''s quality-driven advice, the 'Meta-Expert' dynamically adjusts their relative influence on the final action selection. This dual-perspective approach allows our framework to achieve a near-optimal sampling strategy that minimizes queries while maximizing confidence.

The environment is modeled with a stochastic setting where the loss for each entity $j$ at time step $t$ is constructed from a transformed Bernoulli process. After each action $I_t$, the environment reveals a binary outcome, $r_{t,j} \in \{0,1\}$, where $r_{t,j} = 1$ signifies that entity $j$ was returned by the LLM. Crucially, the environment also provides two measures of uncertainty that modulate this binary outcome: 1) A confidence score, $A_c(I_t,j)$, which reflects the reliability of a positive outcome ($r_{t,j} = 1$), And 2) A query relevance score, $p_{t,k}^{(\text{noise})}$, derived from the sampled observables for the query $I_t$ and can be seen as a realization of $p_g(I_t,j)$ . These components, along with a constant hyperparameter $C_{back}$, which is the hyperparameter reflects the LLM confidence in the absent entities, are combined to form the confirmation-weighted loss that 'DUETS' tracks:

$$\ell(r_{t,j}, A_c(I_t,j), p_{t,k}^{(\text{noise})}; C_{back}) = r_{t,j} \cdot A_c(I_t,j) \ + \ (1-r_{t,j}) \cdot p_{t,k}^{(\text{noise})} \cdot C_{back} \tag{3}$$

Intuitively, when an entity is present ($r_{t,j} = 1$), the loss is determined solely by the confirmation atoms' confidence for positive predictions, penalizing unreliable positives. When the entity is absent, this loss is attenuated by the observation relevance $p_g(I_t,j)$, ensuring that only relevant queries contribute strongly to the framework's statistical engine.

The complete algorithmic details of DUETS are provided in the Supplementary Material Section B. Subsection B.0.3 provides implementation-ready pseudocode with mathematical operations.

### 3.3 Confirmation Atoms: A Dynamic Feedback System

As discussed before, most state-of-the-art methods for ECQA employs additional LLM queries to validate results and assign confidence scores. We abstract these validation routines into a modular structure of *"confirmation atoms(CA)."* As described previously, a central innovation of our framework is the dual purpose these atoms serve. Their primary function is to probe the LLM's output and generate a confidence score for the returned results. This score is the critical signal used by our Statistical Engine to calculate the MAB's intrinsic loss. Their second, novel function, is to provide the *residual information* necessary for the online updating of our framework's internal beliefs about the system. To make this process principled, each atom is designed to probe a distinct source of uncertainty, which we explicitly separate into epistemic (model-based) and aleatoric (data-based) types [Hüllermeier and Waegeman, 2021]. Table 1 summarizes how each atom contributes to the confidence score and which internal components it updates.

| Confirmation Atom | Uncertainty Type | Updates $Mapping$ | Updates $G_t$ | Updates $S$ |
|---|---|---|---|---|
| Counterfactual Agreement | Epistemic | — | ✓ | ✓ |
| Graph Cohesion | Aleatoric | — | ✓ | ✓ |
| The Round-Trip Atom | Epistemic | ✓ | — | ✓ |
| Knowledge Grounding | Epistemic | ✓ | — | ✓ |

Table 1: The relationship between each Confirmation Atom and the framework components it updates. All atoms contribute to the confidence score $A_c(I_t,j)$ which is fed into the Statistical Engine ($S$).

Here we provide a short description of the CAs. The full description of the CAs together with the formal way they update the beliefs are in Supplementary Section C. The Counterfactual Agreement

Atom measures epistemic uncertainty by quantifying the stability of the LLM's predictions when the initial set of observables is perturbed. The Graph Cohesion Atom assesses aleatoric uncertainty by evaluating the semantic plausibility of the returned entities, measuring their average distance within the entity correlation graph. The Round-Trip Atom probes the LLM's internal coherence through a self-consistency check: it first retrieves an entity from a set of observables, then asks the LLM to generate observables for that entity, comparing the initial and final sets. Finally, the Knowledge Grounding Atom provides a direct factual check by comparing the LLM-generated observables for a given entity against a curated, external database. Together, these atoms provide a multi-faceted view of the LLM's output quality, which is aggregated into a single confidence score.

While each confirmation atom provides a distinct signal, a single, unified confidence score is required to drive the updates of the statistical engine. We define the total confidence score $A_c(I_t, j)$ for a returned entity $E_j$ at time step $t$ as a normalized weighted aggregation of the individual atom scores.

First, we transform the Entity Neighborhood Dispersion (END) score, which measures dispersion, into a normalized cohesion score, $\text{Cohesion}_t = 1 - \frac{\text{END}_t}{\max(\text{dist}_{G_t})}$. For each entity $E_j$, the individual atom scores are represented by $\mathbf{u}_{j,t} = [U_A(E_j), U_C(E_j), U_G(E_j), \text{Cohesion}_t]^T$, and their relative importance is defined by a non-negative hyperparameter weight vector, $\mathbf{w} = [w_A, w_{RT}, w_{KG}, w_{GC}]^T$. The final confidence score is then computed as:

$$A_c(I_t, j) = \frac{\mathbf{w} \cdot \mathbf{u}_{j,t}}{\|\mathbf{w}\|_1} \tag{4}$$

where $\|\mathbf{w}\|_1$ is the L1 norm of the weight vector, ensuring the score is a convex combination that remains in the range $[0, 1]$. This normalized score $A_c(I_t, j)$ serves as a single, potent signal that encapsulates the evidence gathered in each trial. It is then fed into the statistical engine to update the total observed count $k_j$ and total expected count $\lambda_j$.

# 4  Evaluations - Parliamentary Work.

Our evaluations are based on the hallmark problem of pathway enrichment analysis, which was described in 1. For this, we collected a corpus of 180 datasets, spanning multiple diseases and conditions, drawn from three related biological benchmarks [Buzzao et al., 2024, Geistlinger et al., 2021, Hutter and Zenklusen, 2018]. Each dataset contains raw gene-expression measurements (features) for control and disease groups, as well as a list of known biological pathways that serve as ground-truth labels associated with diseases. This structure allows us to fully validate our results, and it also used as the prior knowledge required in our framework.

Our evaluations are designed to test three overarching goals: 1) Showing the effectiveness of results aggregating over partial queries. Although it has been shown before, we believe we are the first to use such a comprehensive benchmark. 2) Demonstrating the ARISE effectiveness through token efficiency. 3) Performing a deep ablation study investigating the different parts of ARISE and DUETS, including the "no prior-knowledge" case.

**Replicating the work of Hu et al. [2025b] on our datasets.**  Our first evaluation aims to demonstrate the need for a sophisticated query mechanism such as partition-and-aggregate. We used the annotated corpus described above to perform a large-scale real-data study following the work of Hu et al. [2025b]. As shown in Figure 3 in Supplementary Materiel Section A, even the most advanced models like GPT-4 (more specifically, gpt-4-1106-preview), which was used in the replication of the work of Hu et al. [2025b] on our benchmarks, did not achieve sufficient accuracy. On our corpus of data, a weak association was observed between the model's self-reported confidence and semantic similarity ($r = 0.22$ for Pearson correlation) between the pathways' original names and the names generated by the model, along with a substantial tail of low-similarity predictions.

**Synthetic evaluation of DUETS.**  For evaluating DUETS, we used a controlled synthetic environment that simulates real-world conditions with noisy, graph-structured side observations. This setup allows us to measure DUETS's sample efficiency and its ability to navigate complex dependencies. We created an environment with $K = 60$ actions divided into $C = 3$ clusters, with $m^\star = 2$ relevant actions per cluster, and a *hubbed* feedback graph that controls which side observations are revealed when an action is played (see Supplementary Material Section A). Query quality follows a cluster-aware matrix, so playing an action gives high-quality evidence for nearby entities and low-quality evidence for entities in other clusters. Because hubs reveal more neighbors, we evaluate

rankings using inverse propensity weighting (IPW) to correct for bias. We compare three methods: *GraphOnly*, which explores the feedback graph structure; *NoiseOnly*, which focuses on quality-aware lookahead; and *DUETS*, our approach that mixes both strategies online. As shown in Figure 2 in Supplementary Materiel Section A, DUETS accelerates discovery by combining both sources of information, reaching $80\%$ recall in 375 rounds (median) compared to 390 for NoiseOnly and 428 for GraphOnly. These results show that DUETS learns faster and is more sample-efficient. Its advantage holds compared to the two other methods.

## 5 Conclusions

Our work addresses the critical trade-off between reliability and computational cost in entity-centric question answering (ECQA) from long, complex contexts. Current methods, while effective, often lead to a "token explosion" that renders them impractical for large-scale scientific discovery. To overcome this, we introduced **ARISE**, a novel framework that reframes ECQA as a multi-armed bandit problem with side observations. ARISE's core innovation is the **DUETS Bandit**, a dual-expert online learning algorithm that intelligently synthesizes prior structural knowledge ('GraphExpert') with expected observation quality ('NoiseExpert') to guide an efficient query policy. This is complemented by a modular system of **Confirmation Atoms** for robust, multi-faceted validation and a **Statistical Engine** that moves beyond opaque self-reported scores to provide rigorous, entity-wise p-values under an explicit null hypothesis. Our preliminary results are promising. On synthetic data, DUETS demonstrates superior sample efficiency compared to single-expert policies, confirming the value of its adaptive mixing strategy. Furthermore, our baseline replication on over 180 real-world gene expression datasets highlights the limitations of current single-query approaches.

**Limitations and Future Work.** While ARISE presents a promising direction, we acknowledge several limitations that offer avenues for future research. First, ARISE relays on the availability of a relevant prior knowledge corpus. Although we have outlined a robust "uninformed initialization" protocol, its performance relative to a well-initialized model needs to be thoroughly benchmarked. Second, while ARISE is designed for efficiency, its scalability to extremely large sets of entities (e.g., tens of thousands) has not yet been tested. Finally, our framework assumes that the underlying LLM behaves as a consistent, stateless oracle. The performance of ARISE could be impacted by significant stochasticity in LLM responses or by unannounced updates to proprietary models, which could introduce non-stationarity into the learning environment.

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

 **Technical Appendices and Supplementary Material**

 # A   Evaluation

525 We evaluate along two complementary axes. First, a controlled *synthetic* study that isolates the
526 contribution of the online policy (DUETS) under graph-structured, noisy side-observations. Second,
527 an ongoing *real-data* study that follows the work of Hu et al Hu et al. [2025b] to benchmark ARISE
528 against contemporary LLM-based baselines on annotated gene-expression datasets.

529 ### A.0.1   Synthetic evaluation: DUETS sample efficiency under graph-structured
530 ### side-observations

531 To isolate the contribution of the online policy itself, we benchmark DUETS on a controlled synthetic
532 environment that mirrors the setting in Section 3: actions correspond to entities (pathways), pulling
533 one action reveals *noisy side-observations* about many others, and which observations are revealed is
534 governed by a *feedback graph*.

535 **Environment.**   We simulate $K = 60$ actions partitioned into $C = 3$ clusters of equal size. A small
536 subset of actions are truly relevant: we draw $m^\star = 2$ per cluster (6 in total) and set their Bernoulli
537 success probabilities to $\theta_j = \theta_{\mathrm{hi}} = 0.75$; the remaining actions have $\theta_j = \theta_{\mathrm{lo}} = 0.10$. Querying
538 action $i$ produces a *revealed/hidden* mask according to a directed feedback matrix $P \in [0, 1]^{K \times K}$
539 (row $i$ gives the probability that $j$ is revealed when $i$ is played), and *quality* weights according to
540 $S \in [0, 1]^{K \times K}$ (row $i$ gives the observation quality for all $j$). We instantiate a clustered, **hubbed
541 feedback graph**. In each cluster we designate $25\%$ of actions as *hubs*—actions whose feedback
542 rows have high *out-coverage* (large $\sum_j P_{ij}$), meaning that playing a hub $i$ tends to reveal many
543 neighbors. Concretely, for same-cluster $j$ we set $P_{ij} = 0.95$ if $i$ is a hub and $P_{ij} = 0.12$ if $i$ is a
544 non-hub; cross-cluster reveals are rare with $P_{ij} = 0.01$. Observation quality is high within clusters
545 and low across clusters ($S_{ij} = 0.90$ within, $S_{ij} = 0.12$ across), with small Gaussian jitter (clipped to
546 $[0, 1]$). A single round proceeds as follows: after playing $i$, each $j$ is *revealed* with probability $P_{ij}$; if
547 revealed, we draw $r_{t,j} \sim \mathrm{Bernoulli}(\theta_j)$ and record a reward $r_{t,j} S_{ij}$; otherwise the reward for $j$ is
548 zero. We use the loss $\ell_{t,j} = 1 - r_{t,j} S_{ij}$.

549 **Unbiased ranking via inverse propensity weighting (IPW).**   Because hubs reveal more neighbors,
550 a naïve cumulative-reward ranking is biased. We therefore build, for each policy, a per-arm *IPW*
551 estimator of the latent relevance $r_j$:

$$\widehat{r}_{t,j} \;=\; \sum_{\tau \le t} \frac{\mathrm{obs}_{\tau,j}}{P_{I_\tau j} S_{I_\tau j} + \varepsilon}, \qquad \mathrm{obs}_{\tau,j} = \mathbf{1}\{j \text{ revealed}\} \cdot r_{\tau,j} S_{I_\tau j},$$

552 with a small $\varepsilon$ for numerical stability. This estimator is unbiased for $\mathbb{E}[r_j]$. At round $t$ we rank actions
553 by $\widehat{r}_{t,j}$ and report *Recall@$m^\star$* (the fraction of the $m^\star$ ground-truth actions appearing in the top-$m^\star$
554 estimated list).

555 **Policies.**   We compare three policies; all hyperparameters are identical to the code used to produce
556 Fig. 2.

557 - **GraphOnly.** An Exp3-style learner (following the Exp3 algorithm of Alon et al Alon et al.
558 [2017]) that uses the known feedback graph $P$ to enforce exploration on a dominating set
559 $D_t$ of the current graph. The sampling distribution is $p_t^{\mathrm{graph}} = (1 - \lambda) \frac{w_t}{\|w_t\|_1} + \frac{\lambda}{|D_t|} \mathbf{1}_{D_t}$
560 with $\lambda = 0.35$ and learning rate $\eta_G = 0.25$. We update weights using an *importance-
561 weighted* estimator computed *only* on revealed coordinates: $\widehat{\ell}_{t,j}^{\mathrm{graph}} = \min\{\ell_{t,j}/(P_{I_t j} +$
562 $10^{-12}), \mathrm{cap}\} \cdot \mathbf{1}\{j \text{ revealed}\}$, with a cap of 50 to control variance.

563 - **NoiseOnly.** A quality-aware look-ahead policy that chooses actions expected to yield the
564 most informative side-observations. It maintains an exponential moving average of per-arm
565 rewards, $\widehat{r} \leftarrow (1 - \beta)\widehat{r} + \beta(1 - \ell_t)$ with $\beta = 0.05$, and samples from a softmax over utilities
566 $U_t(i) = \sum_j (S \odot P)_{ij} \widehat{r}_j$ (temperature $1/\eta_N$, with $\eta_N = 1.0$).

567 - **DUETS.** Our meta-learner mixes the two advisers: $p_t = (1 - \alpha_t) p_t^{\mathrm{graph}} + \alpha_t p_t^{\mathrm{noise}}$.
568 During a short *warm-up* of 40 rounds we use a fixed $\alpha_t = \alpha_{\mathrm{warm}} = 0.20$ to en-
569 sure coverage. Thereafter, $\alpha_t$ is learned online by Hedge with meta-rate $\eta_{\mathrm{meta}} = 1.5$:
570 $W_{t+1}^{\mathrm{G}} = W_t^{\mathrm{G}} \exp(-\eta_{\mathrm{meta}} \cdot \langle p_t^{\mathrm{graph}}, \ell_t \rangle)$, $W_{t+1}^{\mathrm{N}} = W_t^{\mathrm{N}} \exp(-\eta_{\mathrm{meta}} \cdot \langle p_t^{\mathrm{noise}}, \ell_t \rangle)$, and

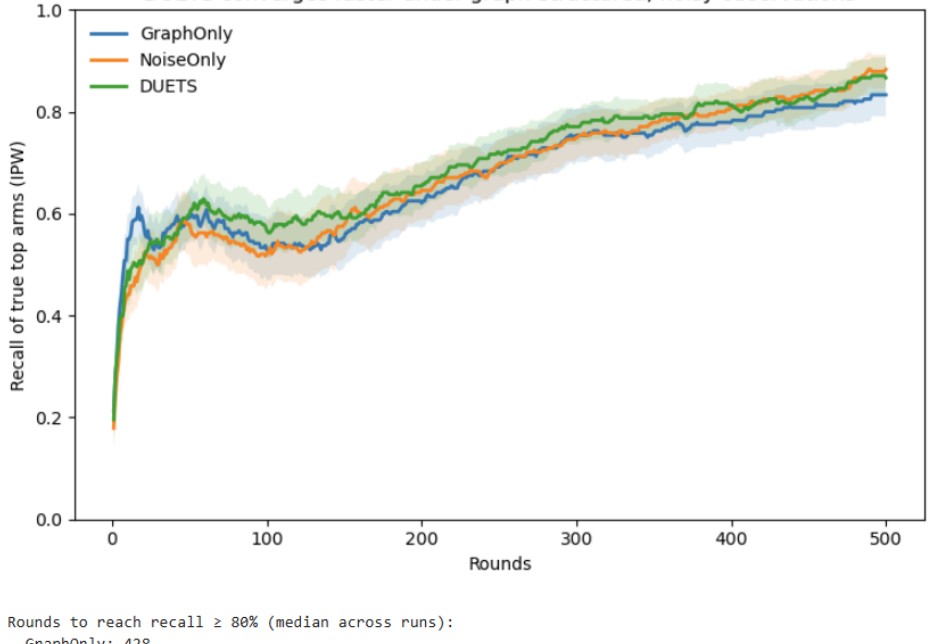

```
Rounds to reach recall ≥ 80% (median across runs):
   GraphOnly: 428
   NoiseOnly: 390
   DUETS    : 375
```

Figure 2: **Synthetic evaluation with a hubbed feedback graph.** Shaded bands are $95\%$ CIs over 40 seeds. We report recall of the true top arms using inverse-propensity weighting (IPW) to debias coverage. DUETS attains $80\%$ recall in 375 rounds (median) versus 390 for `NoiseOnly` and 428 for `GraphOnly`, reflecting faster sample-efficient discovery while maintaining competitive late-round performance.

$\alpha_t = W_t^{\mathrm{N}}/(W_t^{\mathrm{G}} + W_t^{\mathrm{N}})$, with on-the-fly normalization to prevent numeric under/overflow. DUETS uses the same graph and noise sub-learners as above ($\lambda = 0.35$, $\eta_G = 0.25$, $\eta_N = 1.0$, $\beta = 0.05$).

**Protocol and metric.** We run each policy for $T = 500$ rounds on independent environments (40 random seeds) and report the mean recall curve with $95\%$ confidence bands. For a compact sample-complexity summary we also report, for each policy, the median number of rounds needed to reach $\geq 80\%$ Recall@$m^\star$.

**Results.** Figure 2 shows mean recall with $95\%$ CIs over 40 runs (evaluation by inverse-propensity weighting). The hubbed feedback makes graph structure consequential, and IPW removes the coverage bias induced by hubs. In this regime, **DUETS** accelerates early discovery by combining (i) structural coverage from the `GraphOnly` dominating-set exploration and (ii) quality-aware look-ahead from `NoiseOnly`. After a short warm-up, the Hedge meta-update shifts weight toward the stronger adviser online. Quantitatively, DUETS reaches $80\%$ recall in **375** rounds (median), compared to **390** for `NoiseOnly` and **428** for `GraphOnly`; end-of-horizon recall remains competitive across methods.

### A.0.2 Real-data evaluation: Planned ARISE comparison

To assess the performance of ARISE on real data, we compare to recent benchmarks established by Hu et al. Hu et al. [2025b], who evaluated five large language models on the task of assigning functional names to gene sets. In their study, LLMs such as GPT-4 and Gemini Pro were prompted with full lists of genes and tasked with producing a descriptive pathway name together with a self-reported confidence score. GPT-4 was found to generate names similar to curated Gene Ontology (GO) terms in over $70\%$ of cases, with its confidence estimates predictive of correctness; it also showed the strongest ability to decline naming incoherent or random sets, a crucial property for scientific reliability.

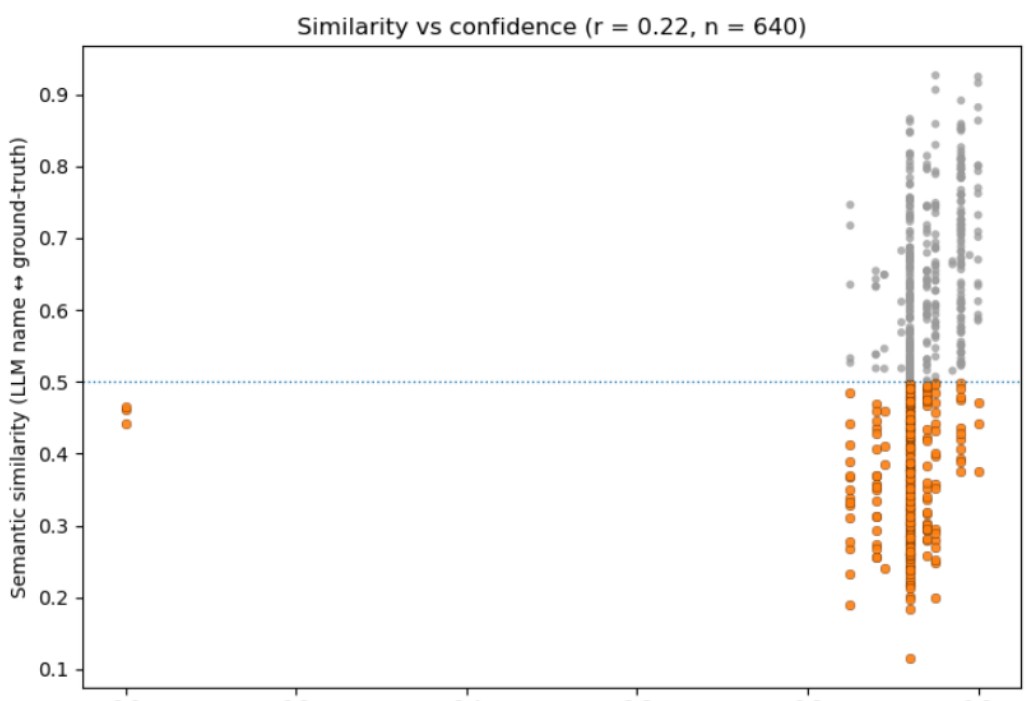

Figure 3: Baseline replication on our 180+ datasets using the Hu et al. pipeline: GPT-4's self-reported confidence versus semantic similarity between the LLM-produced pathway name and the ground-truth pathway name. Points in the lower-right (high confidence, low semantic similarity) indicate likely evaluation mismatches or model overconfidence.

**Our Dataset.** To enable systematic evaluation of ARISE, we assembled a large corpus of more than **180 annotated gene expression datasets**, spanning multiple diseases and experimental conditions. This corpus provides a diverse and challenging benchmark for entity-centric question answering in biology.

**Reproducing the Baseline.** As a first step, we re-implemented the evaluation pipeline from Hu et al., running their published code on our 180+ datasets. This produced baseline results consisting of (i) the pathway names assigned by the LLM to each dataset, and (ii) the model's self-reported confidence scores. These outputs form a direct replication of the Hu et al benchmark, but on a broader and more heterogeneous testbed. As shown in Figure 3, the Pearson correlation between model confidence and the semantic similarity of generated versus ground-truth names is $r = 0.22$ (weak association); moreover, a substantial fraction of generated names have similarity $< 0.5$.

**Planned Comparison with ARISE.** Our next step is to run the ARISE framework incorporating Confirmation Atoms, the DUETS bandit policy, and the statistical significance engine on the same datasets. This will allow a direct, head-to-head comparison between ARISE and the baseline pipeline. We hypothesize that ARISE will outperform the baseline by achieving higher accuracy at substantially lower query cost, while also providing calibrated, interpretable significance estimates rather than opaque self-reported confidence scores.

## B  The DUETS Algorithm: An Adaptive Dual-Perspective Solution

### B.0.1  Motivation: Reconciling Disparate Priors in a Concrete Setting

Our problem is motivated by a concrete scenario: learning which entities are most likely to be returned by a query to a Large Language Model (LLM). In this setting, the true reward $r_{t,j} \in \{0, 1\}$ for an entity $j$ is determined by its absence or presence in the LLM's response. For this we leverage two distinct, independent sources of prior knowledge that an effective learning agent use:

1. **A Graph-Based Co-occurrence Prior:** The literature provides data on the co-occurrence probabilities of different entities. This knowledge is best represented as a directed graph $G_t$, realized from a known probability matrix $P = \{p_{ij}\}$, where an edge suggests a likely co-occurrence. To leverage this, an agent should behave as if it is exploring a sparse, partial-information landscape, where observing one entity provides a strong signal to observe its neighbors. This perspective is directly inspired by the feedback graph model of Mannor and Shamir Mannor and Shamir [2011].

2. **An Observation Quality Prior:** The query mechanism itself introduces another layer of complexity. A query for entity $i$ is performed using a specific set of its "observables" (features). While this provides the best possible observation for entity $i$, the same set of observables also provides a noisy signal about all other entities $j$. The quality of these observations, represented by $p_g(I_t, j)$, is stochastic but drawn from a known distribution. This implies a noisy full-information setting, where the agent's action $I_t$ determines the observation quality for the entire system. This setup shares conceptual similarities with the noisy side-observation models explored by Kocák et al. Kocák et al. [2016].

These two priors suggest fundamentally different algorithmic strategies. The **DUal Experts for Turbid side-Observations with Stochastic feedback graph (DUETS)** algorithm is designed to resolve this tension. It creates a single agent that maintains two parallel worldviews—one partial-information and one full-information—and learns online how to best combine their advice.

### B.0.2 Algorithmic Framework: Adaptive Mixing of Two Expert Perspectives

The 'DUETS' algorithm consists of three core components, each justified by the need to handle a specific aspect of the problem:

- A **GraphExpert**, which operates under the assumption that feedback is sparse and determined by the graph $G_t$. Its purpose is to enforce a robust exploration strategy that respects the co-occurrence prior. Its design is heavily influenced by the 'Exp3.G' family of algorithms from Alon et al. **?**, which demonstrate that leveraging graph structure (e.g., dominating sets) is critical for efficient exploration in partial-information settings.

- A **NoiseExpert**, which acknowledges the noisy full-information reality. Its purpose is to strategically choose an action that maximizes the overall quality of the observations it receives. Unlike the reactive model in Kocák et al. Kocák et al. [2016], where noise quality is unknown and adversarial, our 'NoiseExpert' can be proactive because the statistics of the noise ($\bar{p}_g(I_t, j)$) are known. It performs a lookahead calculation to find the most informative action.

- A high-level **Meta-Expert**, which acts as an adaptive mixer. This is a standard and powerful technique from the "learning from expert advice" literature. Its purpose is to learn the optimal blending of the two sub-experts' advice by tracking their historical performance, thus freeing the user from having to manually set a fixed mixing parameter.

**Consulting the Experts.** The two experts generate their advice independently, based on their distinct worldviews.

- The 'GraphExpert''s distribution, $p_t^{\text{graph}}$, must ensure exploration. Following Alon et al. Alon et al. [2015], an effective strategy is to guarantee a minimum level of exploration on a dominating set $D_t$ of the current graph $G_t$. This ensures that all nodes are observed (in the hypothetical partial-information world) with high probability.

- The 'NoiseExpert''s utility function, $U_t(i)$, is a proactive, one-step lookahead. It estimates the total "information reward" from playing action $i$, weighting the expected quality of each observation $\bar{p}_g(I_t, j)$ by the current estimated reward of action $j$. This prioritizes choosing queries that yield high-quality information about promising entities.

**The Dual Update and its Estimators.** This is the core of the algorithm's dual nature. After observing the outcome, both experts update their internal state, but they interpret the information differently.

- The 'NoiseExpert' uses the simple, low-variance estimator $\tilde{\ell}_{t,k}$. This is possible because it operates in the full-information world and has access to the signal for every action.

- The 'GraphExpert' must use the high-variance, importance-weighted estimator $\hat{\ell}_{t,k}^{\text{graph}}$. The term $\mathbb{I}\{(I_t, k) \in \mathcal{E}_t\}$ enforces its worldview that it only "sees" feedback along realized edges. The denominator $q_{t,k}$ is the probability of this event occurring. Dividing by $q_{t,k}$ is essential to correct for the selection bias and ensure that the estimator is unbiased in expectation ($\mathbb{E}[\hat{\ell}_{t,k}^{\text{graph}}] = \ell_{t,k}$). This importance weighting is a cornerstone of modern bandit algorithms, essential for handling partial feedback as seen in works from Li et al. **?** to Esposito et al. **?**.

**Updating the Meta-Expert.** The 'Meta-Expert' learns by evaluating the advice of its sub-experts in hindsight. The meta-loss, $L_t^{\text{meta,G}}$, represents the expected loss the agent would have suffered if it had followed the 'GraphExpert''s recommendation $p_t^{\text{graph}}$ precisely. By updating its weights based on these meta-losses, the 'Meta-Expert' learns to increase the influence ($\alpha_t$) of the sub-expert that provides consistently better recommendations for the given environment.

### B.0.3 The DUETS Algorithm: Implementation-Level Pseudo-code

This section provides a highly detailed pseudocode for the **DUETS** algorithm, intended to serve as a direct guide for implementation. Each step is broken down into its constituent mathematical and logical operations.

**The Loss Model** The algorithm operates in a full-information setting where, after each round, the true binary outcome $r_{t,j} \in \{0, 1\}$ and the parameters $A_c(t)$ and $p_g(I_t, j)$ are revealed for all entities $j$. The algorithm then constructs the loss for the round using the following function:

$$\ell(r_{t,j}, A_c(I_t, j), p_{t,k}^{(\text{noise})}; C_{back}) = r_{t,j} \cdot A_c(I_t, j) + (1 - r_{t,j}) \cdot p_{t,k}^{(\text{noise})} \cdot C_{back} \tag{5}$$

This constructed loss, which incorporates various measures of uncertainty, is then used to update all expert components.

**Helper Functions** For clarity, we first define two helper functions that will be used within the main algorithm.

---

**Algorithm 1** *

Function `GreedyDominatingSet`$(G = (V, \mathcal{E}))$

1: **Input:** A directed graph $G = (V, \mathcal{E})$.
2: **Initialize:** Dominating set $D \leftarrow \emptyset$, Uncovered nodes $U \leftarrow V$.
3: **while** $U$ is not empty **do**
4:     Let $N_{out}(v) \leftarrow \{v\} \cup \{j \in V \mid (v, j) \in \mathcal{E}\}$.
5:     Select node $v^* \in V$ that maximizes $|N_{out}(v) \cap U|$.
6:     $D \leftarrow D \cup \{v^*\}$.
7:     $U \leftarrow U \setminus N_{out}(v^*)$.
8: **end while**
9: **Return** $D$.

---

**Algorithm 2** *

Function `NormalizeWeights`$(w)$

1: **Input:** A vector of non-negative weights $w = \{w_1, \dots, w_K\}$.
2: $W \leftarrow \sum_{k=1}^{K} w_k$.
3: **if** $W = 0$ **then return** uniform distribution $\{1/K, \dots, 1/K\}$.
4: **elsereturn** $\{w_1/W, \dots, w_K/W\}$.
5: **end if**

---

**Main Algorithm** The main loop of the DUETS algorithm integrates the advice from its three expert components to make decisions and learn from feedback.

---

**Algorithm 3** The DUETS Algorithm (Detailed)

---

**Require:** Set of actions (entities) $V$, $|V| = K$; Number of rounds $T$.
**Require:** Learning rates: $\eta_G, \eta_N, \eta_{meta} > 0$; Regularization parameter $\gamma > 0$.
**Require:** GraphExpert exploration parameter $\lambda_G \in [0, 1]$.
**Require:** Known co-occurrence probability matrix $P \in [0, 1]^{K \times K}$, where $P_{ij} = p_{ij}$.
**Require:** Known constant hyperparameter $a_{cb}$.

1: **Initialize Data Structures:**
2:    GraphExpert weights: $w_1^{\text{graph}} \leftarrow \{1, \ldots, 1\} \in \mathbb{R}^K$.
3:    NoiseExpert weights: $w_1^{\text{noise}} \leftarrow \{1, \ldots, 1\} \in \mathbb{R}^K$.
4:    Meta-Expert weights: $W_1^{\text{meta,G}} \leftarrow 1$, $W_1^{\text{meta,N}} \leftarrow 1$.
5:    Cumulative losses for NoiseExpert's model: $L_0^{\text{noise}} \leftarrow \{0, \ldots, 0\} \in \mathbb{R}^K$.
6:    Running sum for $A_c$: $S_{Ac} \leftarrow 0$; Running count for $A_c$: $N_{Ac} \leftarrow 0$.
7: **for** $t = 1, \ldots, T$ **do**
8:    **Observe Context:** An external process provides the realized graph $G_t = (V, \mathcal{E}_\sqcup)$.
9:      **— Consult GraphExpert —**
10:    Compute dominating set $D_t \leftarrow \texttt{GreedyDominatingSet}(G_t)$.
11:    Normalize weights: $p_t^{\text{w,graph}} \leftarrow \texttt{NormalizeWeights}(w_t^{\text{graph}})$.
12:    Form GraphExpert's mixed distribution for all $k \in V$:
$$p_{t,k}^{\text{graph}} \leftarrow (1 - \lambda_G) \cdot p_{t,k}^{\text{w,graph}} + \frac{\lambda_G}{|D_t|} \cdot \mathbb{I}\{k \in D_t\}.$$
13:      **— Consult NoiseExpert —**
14:    For each pair $(i, j)$, compute the estimated quality: $\hat{p}_g(i, j) \leftarrow$ $\texttt{CalculateExpectedPg}(i, j)$.
15:    Let $\text{est\_reward}_{t,j} \leftarrow 1 - \frac{L_{t-1,j}^{\text{noise}}}{t-1} \cdot \mathbb{I}\{t > 1\}$.
16:    Compute lookahead utilities for all $i \in V$: $U_t(i) \leftarrow \sum_{j=1}^K \text{est\_reward}_{t,j} \cdot \hat{p}_g(i, j)$.
17:    Compute unnormalized weights: $w_{t,k}^{\text{u,noise}} \leftarrow \exp(\eta_N \cdot U_t(k))$.
18:    Normalize to form distribution: $p_t^{\text{noise}} \leftarrow \texttt{NormalizeWeights}(w_t^{\text{u,noise}})$.
19:      **— Consult Meta-Expert and Mix Advice —**
20:    Compute dynamic mixing parameter: $\alpha_t \leftarrow W_t^{\text{meta,N}}/(W_t^{\text{meta,G}} + W_t^{\text{meta,N}})$.
21:    Form the final action distribution for all $k \in V$: $p_{t,k} \leftarrow (1 - \alpha_t) \cdot p_{t,k}^{\text{graph}} + \alpha_t \cdot p_{t,k}^{\text{noise}}$.
22:      **— Act and Observe Feedback —**
23:    Draw action to play: $I_t \sim p_t$.
24:    An external process reveals the true binary outcomes: $\{r_{t,j}\}_{j \in V}$.
25:    An external process reveals the scalar loss parameter: $A_c(I_t, j)$.
26:    An external process reveals the vector of loss parameters: $\{p_g(I_t, j)\}_{j \in V}$.
27:      **— Perform Dual Update —**
28:    For each $j \in V$, construct the loss for the round:
$$\ell_{t,j} \leftarrow A_c(I_t, j) \cdot (r_{t,j}) + (1 - r_{t,j}) \cdot p_{t,k}^{(\text{noise})} \cdot C_{back}.$$
29:    **Update NoiseExpert:**
30:      Update cumulative losses: $L_{t,k}^{\text{noise}} \leftarrow L_{t-1,k}^{\text{noise}} + \ell_{t,k}$ for all $k \in V$.
31:      Update weights: $w_{t+1,k}^{\text{noise}} \leftarrow w_{t,k}^{\text{noise}} \cdot \exp(-\eta_N \cdot \ell_{t,k})$ for all $k \in V$.
32:    **Update GraphExpert:**
33:      Compute observation probabilities for all $k \in V$: $q_{t,k} \leftarrow \sum_{i=1}^K p_{t,i} \cdot p_{ik}$.
34:      Form importance-weighted estimators for all $k \in V$:
$$\hat{\ell}_{t,k}^{\text{graph}} \leftarrow \frac{\ell_{t,k}}{q_{t,k} + \gamma} \cdot \mathbb{I}\{(I_t, k) \in \mathcal{E}_\sqcup\}.$$
35:      Update weights: $w_{t+1,k}^{\text{graph}} \leftarrow w_{t,k}^{\text{graph}} \cdot \exp(-\eta_G \cdot \hat{\ell}_{t,k}^{\text{graph}})$ for all $k \in V$.
36:    **Update Online Learning Model for** $A_c(I_t, j)$**:**
37:      $S_{Ac} \leftarrow S_{Ac} + A_c(I_t, j)$;    $N_{Ac} \leftarrow N_{Ac} + 1$.
38:      **— Update Meta-Expert —**
39:    Compute meta-loss for GraphExpert's advice: $L_t^{\text{meta,G}} \leftarrow \sum_{k=1}^K p_{t,k}^{\text{graph}} \cdot \ell_{t,k}$.
40:    Compute meta-loss for NoiseExpert's advice: $L_t^{\text{meta,N}} \leftarrow \sum_{k=1}^K p_{t,k}^{\text{noise}} \cdot \ell_{t,k}$.
41:    Update meta-weights:
$$W_{t+1}^{\text{meta,G}} \leftarrow W_t^{\text{meta,G}} \cdot \exp(-\eta_{meta} \cdot L_t^{\text{meta,G}}).$$
$$W_{t+1}^{\text{meta,N}} \leftarrow W_t^{\text{meta,N}} \cdot \exp(-\eta_{meta} \cdot L_t^{\text{meta,N}}).$$
42: **end for**

---

### B.0.4 Estimating the Quality Score $p_g(i, j)$

The core motivation is to quantify the relationship between the query action $i$ and the observed entity $j$. Specifically, we want to answer the question: **"If we query the LLM using a set of observables sampled for entity $i$, how much evidence should we expect to see for entity $j$?".** We define this quality score, $p_g(i, j)$, as the expected posterior probability of entity $j$, where the expectation is taken over all the evidence (sets of observables) that a query for entity $i$ is likely to produce. Formally, we want to calculate the expectation:

$$p_g(i, j) = \mathbb{E}_{o \sim P(o|\theta_i)} \left[ P(j \mid o) \right] \tag{6}$$

The direct computation of this expectation is intractable due to the combinatorial explosion in the number of possible observable sets $o$. We therefore turn to an information-theoretic analytical approximation, grounded in Large Deviation Theory(LD-T), for this value.

The core of the approximation is to replace the true expectation over all observable sets, $\mathbb{E}_{o \sim P(\cdot|\theta_i)}[P(j|o)]$, with the posterior evaluated at the mean set of observables, $P(j|\mathbb{E}[o])$. The mean observables from entity $i$, $\mathbb{E}[o]$, is a count vector whose empirical distribution is precisely the mean probability vector $\hat{\theta}_i$.

A key result from Large Deviation Theory (Sanov [1957] Sanov's Theorem states that the probability of observing an empirical distribution $\hat{\theta}'$ from a source $k$ is asymptotically given by $P(\dots) \approx \exp(-n \cdot D_{KL}(\hat{\theta}'||\hat{\theta}_k))$, where $n$ is the number of observables.

## C Confirmation Atoms

Our framework leverages a set of "confirmation atoms" to assign per-entity confidence scores based on LLM output behavior. Each atom is designed to probe a distinct source of uncertainty, which we explicitly separate into two types: *epistemic uncertainty* and *aleatoric uncertainty*. The results from these atoms are aggregated into a single confidence score, $A_c(I_t, j)$, for each returned entity $E_j$ at time step $t$.

Here we provide an full description of the CAs.

**1. Counterfactual Agreement Atom**    This atom measures epistemic uncertainty by quantifying the stability of the LLM's predictions under input perturbations. Given an initial observations subset $O_{\text{query}}$, we generate $n$ perturbed queries $\{O_k\}_{k=1}^n$ from neighbored entities from the graph $G_t$ and observe the resulting LLM responses $\{E_{\text{response},k}\}_{k=1}^n$. The Counterfactual Agreement Score $A(E_j)$ for a returned entity $E_j$ is defined as the proportion of perturbed queries that still include $E_j$ in their top predictions:

$$A(E_j) = \frac{1}{n} \sum_{k=1}^n \mathbb{I}[E_j \in E_{\text{response},k}]$$

A low score indicates instability in the prediction, suggesting that the LLM lacks consistent internal knowledge.

**2. Graph Cohesion Atom**    This atom measures aleatoric uncertainty by evaluating the domain plausibility of the LLM's output. It computes an Entity Neighborhood Dispersion (END) score based on the shortest-path distances between the entities returned by the LLM in our a-priori correlation graph $G_t$. Let $\{E_1, \dots, E_k\}$ be the set of entities returned in a trial. The END score is defined as the average pairwise shortest-path distance:

$$\text{END} = \frac{1}{\binom{k}{2}} \sum_{j<m} \text{dist}_{G_t}(E_j, E_m)$$

A low END score indicates a dense, localized cluster of entities, reflecting aleatoric uncertainty—multiple plausible domain interpretations of the same observations subset.

**3. The Round-Trip Atom**    This atom provides a powerful measure of the LLM's internal knowledge coherence. It performs a round-trip verification by first retrieving an entity from a given observations set and then immediately asking the LLM to generate observations for that retrieved entity.

    1. **Forward Pass:** A query with an observations set $O_{\text{query}}$ yields a primary response entity $E_j$.

2. **Reverse Pass:** A second query, "Given entity $E_j$, what are its top $N$ observations?", yields a new observations set $O_{\text{reverse}}$.

The Self-Consistency Score $U_C(E_j)$ is defined as the Jaccard similarity between the initial and reverse-pass observations sets:

$$U_C(E_j) = \frac{|O_{\text{query}} \cap O_{\text{reverse}}|}{|O_{\text{query}} \cup O_{\text{reverse}}|}$$

A high $U_C(E_j)$ indicates robust, self-consistent knowledge.

**4. Knowledge Grounding Atom**   This atom directly addresses factual inconsistency by comparing the LLM's knowledge to an authoritative, external source. It builds upon the Round-Trip Atom, using the observations list $O_{\text{reverse}}$ produced by the LLM. An external query is issued to a curated database to obtain a "ground truth" observations list, $O_{\text{external}}$, for entity $E_j$. The Grounding Score $U_G(E_j)$ is the Jaccard similarity between the two lists:

$$U_G(E_j) = \frac{|O_{\text{reverse}} \cap O_{\text{external}}|}{|O_{\text{reverse}} \cup O_{\text{external}}|}$$

A high $U_G(E_j)$ provides a strong signal of factual accuracy, contributing to the confidence score.

# D Framework Robustness: Uninformed Initialization

A key strength of the **ARISE** framework is its robustness and adaptability, allowing it to function effectively even in the absence of a pre-existing, curated corpus for generating prior knowledge. We address this **uninformed initialization** scenario through three complementary mechanisms.

First, in a practical application where no corpus is available, the framework can use the LLM itself to generate a preliminary set of priors. By prompting the LLM with randomly sampled sets of observables, we can build an initial, albeit noisy, estimate of entity co-occurrence probabilities and observable-to-entity mappings. This serves as a functional starting point for the framework.

More fundamentally, the framework is designed to learn and refine these priors **online** as a core part of its operation. The residual information gathered by the **Confirmation Atoms** is not only used for scoring but also for updating ARISE's internal beliefs. For instance, the **Graph Cohesion Atom** provides direct evidence for updating the stochastic feedback graph, allowing the framework to bootstrap and continuously improve its own knowledge base from the LLM's responses.

Finally, ARISE remains viable even in the most extreme case, assuming no initial priors are provided and the Confirmation Atom updates are disabled.

1. A **feedback graph** is inherently constructed from the very first query. Each list of entities returned by the LLM is a direct observation of their co-occurrence, providing an immediate, dynamically updated graph for the 'GraphExpert' to leverage.

2. The statistical engine remains well-defined. The success probabilities $\{p_i\}$ used to parameterize the **Poisson Binomial distribution** for the null hypothesis would default to a **uniform distribution** over all entities. While uninformative, this is not a misspecification but rather the correct assumption when no relationship between observables and entities is known *a priori*.

3. The **DUETS bandit** is designed to adapt to this uncertainty. Initially, the 'NoiseExpert' (which relies on observable-entity mappings) will provide poor advice. However, the 'MetaExpert' will quickly learn to down-weight its recommendations and rely more heavily on the 'GraphExpert', which learns from the dynamically observed co-occurrence graph. This results in a less sample-efficient "warm-up" period, but the system is designed to converge and find the correct signal.

To validate these claims, we will include a dedicated **ablation study** in our final evaluation to empirically demonstrate the framework's performance under this challenging uninformed initialization scenario.

