# OpenReview forum: "Taming the Noisy Oracle: Robust Entity-Centric Question Answering via Learning from Imperfect Feedback"
_NeurIPS.cc/2025/Workshop/Reliable_ML — NeurIPS 2025 - Reliable ML Workshop_

### Official Review · Reviewer_MPrb · 2025-09-19
**Accept**

**Rating:** 6
**Confidence:** 3

**Review:**

The paper targets entity-centric QA (ECQA) where an LLM acts as a noisy, black-box oracle over long, heterogeneous inputs. It proposes ARISE, a framework that treats each query as a biased side-observation and orchestrates iterative retrieval with a statistical engine for online validation. Central is DUETS Bandit-a dual-expert policy combining a GraphExpert (stochastic feedback graph from priors) and a NoiseExpert (query selection for higher-quality observations), plus modular Confirmation Atoms that produce confidence scores and residual signals for belief updates; the system outputs formal p-values and aims to reduce query complexity. Early synthetic results are promising; a large bio pathway enrichment (PEA) evaluation is ongoing.

## Strengths:
- Compelling vision: Treat LLM feedback as imperfect side-observations and leverage priors + adaptive sampling + explicit validation—coherent and well-motivated for reliability with messy inputs.
- Algorithmic structure: DUETS’ dual experts with a meta-expert mixer is a plausible route to cost-aware robustness in ECQA.
- Validation design: Confirmation Atoms (epistemic/aleatoric) and their aggregation into a single confidence signal provide a principled interface to the statistical engine.
- Relevance: Focus on long, information-dense inputs and the reliability/cost trade-off is squarely in scope for the workshop.

## Weaknesses:
- Empirics immature: Real-world PEA results are in progress; current evidence is primarily synthetic
- Assumption load: Effectiveness depends on prior knowledge graphs and quality of the confirmation process; scalability to very large entity sets is not yet demonstrated.
- Statistical detail: Multiple-testing control, calibration under LLM non-stationarity, and robustness to shifting prompt distributions need deeper treatment.

## Suggestions:
- Complete the PEA evaluation with cost-vs-quality profiles; include strong baselines (partition-query-aggregate; consensus; confidence-scored schemes).
- Provide ablations isolating GraphExpert vs NoiseExpert and sensitivity to prior graph quality; include implementation-ready pseudocode (you reference it).
- Expand the statistical engine section: null construction, p-value calibration, sequential testing control.
- Stress-test scalability (e.g., 10k+ entities) and robustness to LLM drift; discuss lightweight re-calibration strategies.